# HIERARCHICAL FEW-SHOT IMITATION WITH SKILL TRANSITION MODELS

## ABSTRACT

A desirable property of autonomous agents is the ability to both solve long-horizon problems and generalize to unseen tasks. Recent advances in data-driven skill learning have shown that extracting behavioral priors from offline data can enable agents to solve challenging long-horizon tasks with reinforcement learning. However, generalization to tasks unseen during behavioral prior training remains an outstanding challenge. To this end, we present Few-shot Imitation with Skill Transition Models (FIST), an algorithm that extracts skills from offline data and utilizes them to generalize to unseen tasks given a few downstream demonstrations. FIST learns an inverse skill dynamics model, a distance function, and utilizes a semi-parametric approach for imitation. We show that FIST is capable of generalizing to new tasks and substantially outperforms prior baselines in navigation experiments requiring traversing unseen parts of a large maze and 7-DoF robotic arm experiments requiring manipulating previously unseen objects in a kitchen.

## 1 INTRODUCTION

We are interested in developing control algorithms that enable robots to solve complex and practical tasks such as operating kitchens or assisting humans with everyday chores at home. There are two general characteristics of real-world tasks – long-horizon planning and generalizability. Practical tasks are often long-horizon in the sense that they require a robot to complete a sequence of subtasks. For example, to cook a meal a robot might need to prepare ingredients, place them in a pot, and operate the stove before the full meal is ready. Additionally, in the real world many tasks we wish our robot to solve may differ from tasks the robot has completed in the past but require a similar skill set. For example, if a robot learned to open the top cabinet drawer it should be able to quickly adapt that skill to open the bottom cabinet drawer. These considerations motivate our research question: *how can we learn skills that enable robots to generalize to new long-horizon downstream tasks?*

Recently, learning data-driven behavioral priors has become a promising approach to solving long-horizon tasks. Given a large unlabeled offline dataset of robotic demonstrations solving a diverse set of tasks this family of approaches Singh et al. (2020); Pertsch et al. (2020); Ajay et al. (2021) extract behavioral priors by fitting maximum likelihood expectation latent variable models to the offline dataset. The behavioral priors are then used to guide a Reinforcement Learning (RL) algorithm to solve downstream tasks. By selecting skills from the behavioral prior, the RL algorithm is able to explore in a structured manner and can solve long-horizon navigation and manipulation tasks. However, the generalization capabilities of RL with behavioral priors are limited since a different RL agent needs to be trained for each downstream task and training each RL agent often requires millions of environment interactions.

On the other hand, few-shot imitation learning has been a promising paradigm for generalization. In the few-shot imitation learning setting, an imitation learning policy is trained on an offline dataset of demonstrations and is then adapted in few-shot to a downstream task Duan et al. (2017). Few-shot imitation learning has the

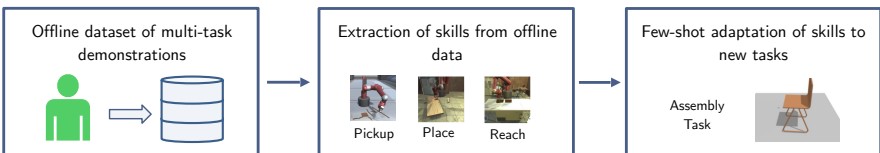

Figure 1: In this work we are interested in enabling autonomous robots to solve complex long-horizon tasks that were unseen during training. To do so, we assume access to a large multi-task dataset of demonstrations, extract skills from the offline dataset, and adapt those skills to new tasks that were unseen during training.

added advantage over RL in that it is often easier for a human to provide a handful of demonstrations than it is to engineer a new reward function for a downstream task. However, unlike RL with behavioral priors, few-shot imitation learning is most often limited to short-horizon problems. The reason is that imitation learning policies quickly drift away from the demonstrations due to error accumulation Ross et al. (2011b), and especially so in the few-shot setting when only a handful of demonstrations are provided.

While it is tempting to simply combine data-driven behavioral priors with few-shot imitation learning, it is not obvious how to do so since the two approaches are somewhat orthogonal. Behavioral priors are trained on highly multi-modal datasets such that a given state can correspond to multiple skills. Given a sufficiently large dataset of demonstrations for the downstream task the imitation learning algorithm will learn to select the correct mode. However, in the few-shot setting how do we ensure that during training on downstream data we choose the right skill? Additionally, due to the small sample size and long task horizon it is highly likely that a naive imitation learning policy will drift from the few-shot demonstrations. How do we prevent the imitation learning policy from drifting away from downstream demonstrations?

The focus of our work is the setup illustrated in Figure 1; we introduce Few-Shot Imitation Learning with Skill Transition Models (FIST), a new algorithm for few-shot imitation learning with skills that enables generalization to unseen but semantically similar long-horizon tasks to those seen during training. Our approach addresses the issues with skill selection and drifting in the few-shot setting with two main components. First, we introduce an inverse skill dynamics model that conditions the behavioral prior not only on the current state but also on a future state, which helps FIST learn uni-modal future conditioned skill distribution that can then be utilized in few-shot. The inverse skill model is then used as a policy to select skills that will take the agent to the desired future state. Second, we train a distance function to find the state for conditioning the inverse skill model during evaluation. By finding states along the downstream demonstrations that are closest to the current state, FIST prevents the imitation learning policy from drifting. We show that our method results in policies that are able to generalize to new long-horizon downstream tasks in navigation environments and multi-step robotic manipulation tasks in a kitchen environment. To summarize, we list our three main contributions:

1. We introduce FIST - an imitation learning algorithm that learns an inverse skill dynamics model and a distance function that is used for semi-parametric few-shot imitation.

2. We show that FIST can solve long-horizon tasks in both navigation and robotic manipulation settings that were unseen during training and outperforms previous behavioral prior and imitation learning baselines.

3. We provide insight into how different parts of the FIST algorithm contribute to final performance by ablating different components of our method such as future conditioning and fine-tuning on downstream data.

## 2 RELATED WORK

Our approach combines ingredients from imitation learning and skill extraction to produce policies that can solve long-horizon tasks and generalize to tasks that are out of distribution but semantically similar to those encountered in the training set. We cover the most closely related work in imitation learning, skill extraction, and few-shot generalization.

**Imitation Learning**: Imitation learning is a supervised learning problem where an agent extracts a policy from a dataset of demonstrations Billard et al. (2008); Osa et al. (2018). The two most common approaches to imitation are Behavior Cloning Pomerleau (1988); Ross et al. (2011a) and Inverse Reinforcement Learning (IRL) Ng & Russell (2000). BC approaches learn policies $\pi_\theta(a|s)$ that most closely match the state-conditioned action distribution of the demonstration data. IRL approaches learn a reward function from the demonstration data assuming that the demonstrations are near-optimal for a desired task and utilize Reinforcement Learning to produce policies that maximize the reward. For simplicity and to avoid learning a reward function, in this work we aim to learn generalizable skills and using the BC approach. However, two drawbacks of BC are that the imitation policies require a large number of demonstrations and are prone to drifting away from the demonstration distribution during evaluation due to error accumulation Ross et al. (2011b). For this reason, BC policies work best when the time-horizon of the task is short.

**Skill Extraction with Behavioral Priors**: Hard-coding prior knowledge into a policy or dynamics model has been considered as a solution to more sample efficient learning, especially in the context of imitation learning Chatzilygeroudis et al. (2019); Bahl et al. (2020); Stulp & Sigaud (2013). For example Stulp & Sigaud (2013) utilizes dynamic movement primitives instead of the raw action space to simplify the learning problem and improve the performance of evolutionary policy search methods. While imposing the structural prior on policy or dynamic models can indeed improve few-shot learning, our method is complementary to these works and proposes to learn behavioral priors from a play dataset. Methods that leverage behavioral priors utilize offline datasets of demonstrations to bias a policy towards the most likely skills in the datasets. While related closely to imitation learning, behavioral priors have been mostly applied to improve Reinforcement Learning. Behavioral priors learned through maximum likelihood latent variable models have been used for structured exploration in RL Singh et al. (2020), to solve complex long-horizon tasks from sparse rewards Pertsch et al. (2020), and regularize offline RL policies Wu et al. (2019); Peng et al. (2019); Nair et al. (2020). While impressive, RL with data-driven behavioral priors does not generalize to new tasks efficiently, often requiring millions of environment interactions to converge to an optimal policy for a new task.

**Few-Shot Learning**: Few-shot learning Wang et al. (2020) has been studied in the context of image recognition Vinyals et al. (2016); Koch et al. (2015), reinforcement learning Duan et al. (2016), and imitation learning Duan et al. (2017). In the context of reinforcement and imitation learning, few-shot learning is often cast as a meta-learning problem Finn et al. (2017); Duan et al. (2016; 2017), with often specialized datasets of demonstrations that labeled by tasks. However, there are other means of attaining few-shot generalization and adaptation that do not require such expensive sources of data. For instance, Cully et al. (2015) and Pautrat et al. (2018) use Bayesian Optimization for transfer learning to a possibly damaged robot in real-world via learned priors based on big repertoires of controllers in different settings from simulation. However, our problem of interest is skill adaptation that requires no further interaction with the downstream environment during few-shot imitation.

Recently, advances in unsupervised representation learning in natural language processing Radford et al. (2019); Brown et al. (2020) and vision He et al. (2020); Chen et al. (2020) have shown how a network pre-trained with a self-supervised objective can be finetuned or adjusted with a linear probe to generalize in few-shot or even zero-shot Radford et al. (2021) to a downstream task. Our approach to few-shot imitation learning is loosely inspired by the generalization capabilities of networks pre-trained with unsupervised objectives. Our approach first fits a behavioral prior to an *unlabeled* offline dataset of demonstrations to extract skills and then fits an imitation learning policy over the previously acquired skills to generalize in few-shot to new tasks.

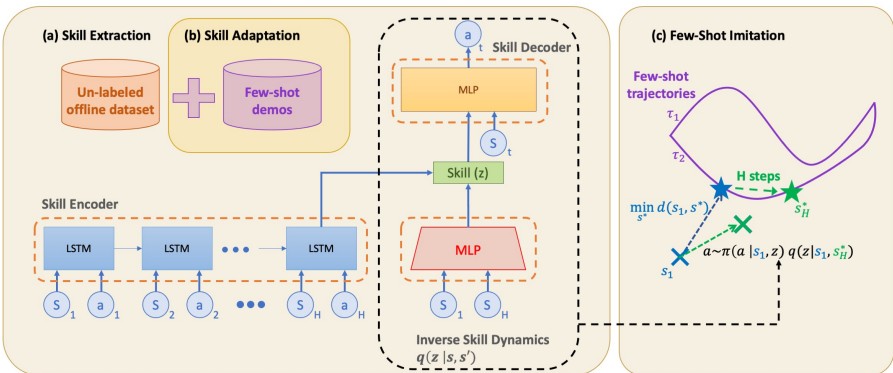

Figure 2: Our algorithm – Few-Shot Imitation Learning with Skill Transition Models (FIST) – is composed of three parts: (a) *Skill Extraction:* we fit a skill encoder, decoder, inverse skill dynamics model, and a distance function to the offline dataset; (b) *Skill Adaptation:* For downstream task, we are given a few demonstrations and adapt the skills learned in (a), by fine-tuning the encoder, decoder, and the inverse model. (c) *Few-Shot Imitation:* finally, to imitate the downstream demonstrations, we utilize the distance function to perform a look ahead along the demonstration to condition the inverse model and decode an action.

# 3 APPROACH

## 3.1 PROBLEM FORMULATION

**Few-shot Imitation Learning**: We denote a demonstration as a sequence of states and actions: $\tau = \{s_1, a_1, s_2, a_2, \ldots, s_T, a_T\}$. In a few-shot setting we assume access to a small dataset of $M$ such expert demonstrations $\mathcal{D}^{\text{demo}} = \{\tau_i\}_{i=1}^{i=M}$ that fulfill a specific long horizon task in the environment. For instance a sequence of sub-tasks in a kitchen environment such as moving the kettle, turning on the burner and opening a cabinet door. The goal is to imitate this behavior using only a few available example trajectories.

**Skill Extraction**: In this work we assume access to an unlabeled offline dataset of prior agent interactions with the environment in the form of $N$ reward-free trajectories $\{\tau_i = \{(s_t, a_t)\}_{t=1}^{t=T_i}\}_{i=1}^{i=N}$. We further assume that these trajectories include semantically meaningful skills that are composable to execute long horizon tasks in the environment. This data can be collected from past tasks that have been attempted, or be provided by human-experts through teleoperation Zhang et al. (2018).

Skill extraction refers to an unsupervised learning approach that utilizes this reward-free and task-agnostic dataset to learn a skill policy in form of $\pi_\theta(a|s, z)$ where $a$ is action, $s$ is the current state, and $z$ is the skill. Our hypothesis is that by combining these skill primitives we can solve semantically similar long-horizon tasks that have not directly been seen during the training. In this work we propose a new architecture for skill extraction based on continuous latent variable models that enables a semi-parametric evaluation procedure for few-shot imitation learning.

## 3.2 HIERARCHICAL FEW-SHOT IMITATION WITH SKILL TRANSITION MODELS

Our method, shown in Fig. 2, has three components: (i) Skill extraction, (ii) Skill adaptation via fine-tuning on few-shot data, and (iii) Evaluating the skills using a semi-parametric approach to enable few-shot imitation.

**(i) Skill Extraction from Offline Data:** We define a continuous skill $z_i \in \mathcal{Z}$ as an embedding for a sequence of state-action pairs $\{s_t, a_t, \ldots, s_{t+H-1}, a_{t+H-1}\}$ with a fixed length H. This temporal abstraction of skills has proven to be useful in prior work Pertsch et al. (2020); Ajay et al. (2021), by allowing a hierarchical decomposition of skills to achieve long horizon downstream tasks. To learn the latent space $\mathcal{Z}$ we propose training a continuous latent variable model with the encoder as $q_\phi(z|s_t, a_t, \ldots, s_{t+H-1}, a_{t+H-1})$ and the decoder as $\pi_\theta(a|s, z)$. The encoder outputs a distribution over the latent variable $z$ that best explains the variation in the state-action pairs in the sub-trajectory.

The encoder is an LSTM that takes in the sub-trajectory of length $H$ and outputs the parameters of a Gaussian distribution as the variational approximation over the true posterior $p(z|s_t, a_t, \ldots, s_{t+H-1}, a_{t+H-1})$. The decoder is a policy that maximizes the log-likelihood of actions of the sub-trajectory conditioned on the current state and the skill. We implement the decoder as a feed-forward network which takes in the current state $s_t$ and the latent vector $z$ and regresses the action vector directly. This architecture resembles prior works on skill extraction Pertsch et al. (2020).

To learn parameters $\phi$ and $\theta$, we randomly sample batches of $H$-step continuous sub-trajectories from the training data $\mathcal{D}$ and maximize the evidence lower bound (ELBO):

$$\log p(a_t|s_t) \geq \mathbb{E}_{\tau \sim \mathcal{D}, z \sim q_\phi(z|\tau)}[\underbrace{\log \pi_\theta(a_t|s_t, z)}_{\mathcal{L}_{\text{rec}}} + \beta \underbrace{(\log p(z) - \log q_\phi(z|\tau))}_{\mathcal{L}_{\text{reg}}}] \tag{1}$$

where the posterior $q_\phi(z|\tau)$ is regularized by its Kullback-Leibler (KL) divergence from a unit Gaussian prior $p(z) = \mathcal{N}(0, I)$ and $\beta$ is a parameter that tunes the regularization term Higgins et al. (2016).

To enable quick few-shot adaptation over skills we learn an inverse skill dynamics model $q_\psi(z|s_t, s_{t+H-1})$ that infers which skills should be used given the current state and a future state that is $H$ steps away. To train the inverse skill dynamics model we minimize the KL divergence between the approximated skill posterior $q_\phi(z|\tau)$ and the output of the state conditioned skill prior. This will result in minimizing the following loss with respect to the parameters $\psi$:

$$\mathcal{L}_{\text{prior}}(\psi) = \mathbb{E}_{\tau \sim \mathcal{D}}\left[D_{KL}(q_\phi(z|\tau), q_\psi(z|s_t, s_{t+H-1}))\right]. \tag{2}$$

We use a reverse KL divergence to ensure that our inverse dynamics model has a broader distribution than the approximate posterior to ensure mode coverage Bishop (2006). In our implementation we use a feed-forward network that takes in the concatenation of the current and future state and outputs the parameters of a Gaussian distribution over $z$. Conditioning on the future enables us to make a more informative decision on what skills to execute which is a key enabler to few-shot imitation. We jointly optimize the skill extraction and inverse model with the following loss:

$$\mathcal{L}(\phi, \theta, \psi) = \mathcal{L}_{\text{rec}}(\phi, \theta) + \beta \mathcal{L}_{\text{reg}}(\phi) + \mathcal{L}_{\text{prior}}(\psi) \tag{3}$$

**(ii) Skill Adaption via Fine-tuning on Downstream Data** : To improve the consistency between the unseen downstream demonstrations and the prior over skills, we use the demonstrations to fine-tune the parameters of the architecture by taking gradient steps over the loss in Equation 3. In the experiments we ablate the performance of FIST with and without fine-tuning to highlight the differences.

**(iii) Semi-parametric Evaluation for Few-shot Imitation Learning**: To run the agent, we need to first sample a skill $z \sim q_\psi(z|s_t, s^*_{t+H})$ based on the current state and the future state that it seeks to reach. Then, we can use the low-level decoder $\pi(a_t|z, s_t)$ to convert that sampled skill $z$ and the current state $s_t$ to the corresponding action $a_t$. During evaluation we use the demonstrations $\mathcal{D}^{\text{demo}}$ to decide which state to use as

the future state to condition on. For this purpose we use a learned distance function $d(s, s')$ to measure the distance between the current state $s_t$ and every other state in the demonstrated trajectories. Then, from the few-shot data we find the closest state $s_t^*$ to the current state according to the distance metric:

$$s_t^* = \min_{s_{ij} \in \mathcal{D}^{\text{demo}}} d(s_t, s_{ij}) \tag{4}$$

where $s_{ij}$ is the $j^{\text{th}}$ state in the $i^{\text{th}}$ trajectory in $\mathcal{D}^{\text{demo}}$. We then condition the inverse dynamics model on the current state $s_t$ and the state $s_{t+H}^*$, $H$ steps ahead of $s_t^*$, within the trajectory that $s_t^*$ belongs to. If by adding $H$ steps we reach the end of the trajectory, we use the end state within the trajectory as the target future state. The reason for this look-ahead adjustment is to ensure that the sampled skill always makes progress towards the future states of the demonstration. After the execution of action $a_t$ according to the low-level decoder, the process is repeated until the fulfillment of the task. The procedure is summarized in Algorithm 1.

---

**Algorithm 1** FIST: Evaluation Algorithm

---

1: **Inputs:** Fine-tuned inverse skill dynamics model $q_\psi(z|s_t, s_{t+H-1})$, fine-tuned skill policy $\pi_\theta(a|s, z)$, learned distance function $d(s, s')$, downstream demonstration $\mathcal{D}^{\text{demo}}$
2: Initialize the environment to $s_0$
3: **for** each $t = [1 \ldots T]$ **do**
4:     Pick $s_t^{*'} = \text{LookAhead}(\min_{s \in \mathcal{D}^{\text{demo}}} d(s_t, s))$
5:     Sample skill $z \sim q_\psi(z|s_t, s_t^{*'})$
6:     Sample action $a \sim \pi_\theta(a|s_t, z)$
7:     $s_t \leftarrow \text{env.step}(a)$

---

### 3.3 LEARNING THE DISTANCE FUNCTION

To enable few-shot imitation learning we use a learned distance function to search for a goal state that is "close" to the current state of the agent. Learning contrastive distance metrics has been successfully applied in prior work both in navigation and manipulation experiments. Both Liu et al. (2020) and Emmons et al. (2020) utilize contrastive distance functions to build semi-parametric topological graphs, and Savinov et al. (2018) and Shah et al. (2020) use a similar approach for visual navigation. Inspired by the same idea, we also use a contrastive loss such that states that are $H$ steps in the future are close to the current state while all other states are far. We refer the reader to the Appendix A.1 for further details.

## 4 EXPERIMENTS

In the experiments we are interested in answering the following questions: (i) Can our method successfully imitate unseen long-horizon downstream demonstrations? (ii) What is the importance of semi-parametric approach vs. future conditioning? (iii) Is pre-training and fine-tuning the skill embedding model necessary for achieving high success rate?

### 4.1 ENVIRONMENTS

We evaluate the performance of FIST on two simulated navigation environments and a robotic manipulation task from the D4RL benchmark as shown in Figure 3. To ensure generalizability to out-of-distribution tasks we remove some category of trajectories from the offline data and at test-time, we see if the agent can generalize to those unseen trajectories. Specifically for Pointmass and Ant environments we block some

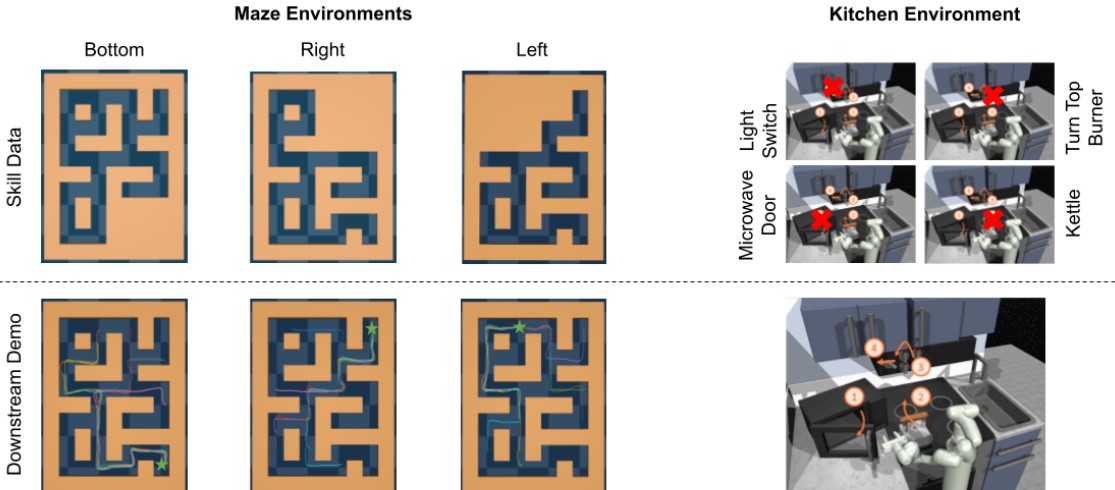

Figure 3: **Top**: In each environment, we block some part of the environment and collect task-agnostic and reward-free trajectories for extracting skills. In the kitchen environment, red markers indicate the objects that are excluded. **Bottom**: For downstream demonstrations, we use 10 expert trajectories that involve unseen parts of the maze or manipulation of unseen objects.

regions of the maze and at test-time we provide demonstration that move into the excluded region; and for Kitchen we exclude interactions with a few selected objects and at test-time we provide demonstrations that involve manipulating the excluded object. Further details on the environment setup and data collection procedure is provided in appendix B.

## 4.2 RESULTS

We use the following approaches for comparison: **BC+FT**: Trains a behavioral cloning agent (i.e. $\pi_\theta(a|s)$) on the offline dataset $\mathcal{D}$ and fine-tunes to the downstream dataset $\mathcal{D}^{demo}$. **SPiRL**: This is an extension of the existing skill extraction methods to imitation learning over skill space Pertsch et al. (2020). The skill extraction method in SPiRL Pertsch et al. (2020) is very similar to FIST, but instead of conditioning the skill prior on the future state it only uses the current state. To adapt SPiRL to imitation learning, after pre-training the skills module, we fine-tune it on the downstream demonstrations $\mathcal{D}^{demo}$ (instead of finetuning with RL as proposed in the original paper). After fine-tuning we execute the skill prior for execution. **FIST (ours)**: This runs our semi-parametric approach after learning the future conditioned skill prior. After extracting skills from $\mathcal{D}$ we fine-tune the parameters on the downstream demonstrations $\mathcal{D}^{demo}$ and perform the proposed semi-parametric approach for evaluation.

Figure 4 compares the normalized average score on each unseen task from each domain. Each tick on x-axis presents either an excluded region (for Maze) or an excluded object (for Kitchen). The un-normalized scores are included in tables 6 and 7 in Appendix C.1. Here, we provide a summary of our key findings [1]:

(i) In the PointMaze environment, FIST consistently succeeds in navigating the point mass into all three goal locations. The skills learned by SPiRL fail to generalize when the point mass falls outside of the training distribution, causing it to get stuck in corners. While BC+FT also solves the task frequently in the Left and Bottom goal location, the motion of the point mass is sub-optimal, resulting in longer episode lengths.

---

[1]The video of some of our experiments are included in the supplementary material.

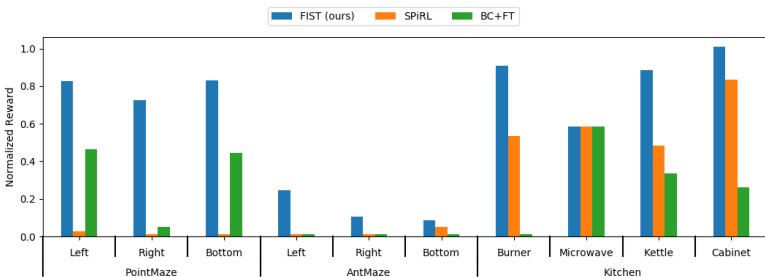

Figure 4: Normalized Reward on all of our environments, and their excluded regions / objects. For maze, the episode length is subtracted from the maximum episode length, which is then divided by the maximum episode length. For kitchen, the reward is the number of sub-tasks completed in order, and normalized by the maximum of 4.0

(ii) In the AntMaze environment, FIST achieves the best performance compared to the baselines. SPiRL and BC+FT make no progress in navigating the agent towards the goal while FIST is able to frequently reach the goals in the demonstrated trajectories. We believe that the low success rate numbers in this experiment is due to the low quality of trajectories that exist in the offline skill dataset $\mathcal{D}$. In the dataset, we see many episodes with ant falling over, and FIST's failure cases also demonstrate the same behavior, hence resulting in a low success rate. We hypothesize that with a better dataset FIST can achieve a higher success rate number.

(iii) In the kitchen environment, FIST significantly outperforms SPiRL and BC+FT. FIST can successfully complete **3 out of 4** long-horizon object manipulation tasks in the same order required by the task. In one of these long-horizon tasks all algorithms perform similarly poor. We believe that such behavior is due to the fact that the majority of the trajectories in the pre-training dataset start with a microwave task and removing this sub-task can substantially decrease the diversity and number of trajectories seen during pre-training.

## 4.3 ABLATION STUDIES

In this section we study different components of the FIST algorithm to provide insight on the contribution of each part. We provide further ablations in Appendix C.2.

**Effect of the Semi-parameteric evaluation and future conditioning of the skill prior.** We recognize two major differences between FIST and SPiRL: (1) future conditioning of the skill prior during skill extraction (2) the semi-paramteric approach for picking the future state to condition on. In order to separately investigate the contribution of each part we performed an ablation with the following modifications of FIST and SPiRL on the kitchen environment: **FIST (Euc.)** is similar to our FIST experiment, except that we use the Euclidean distance on the raw state space, rather than our learned contrastive distance for lookup. Comparison to this baseline measure the importance of our contrastive distance method, compared to a simpler alternative; **SPiRL (closest)** uses the contrastive distance function to look-up the closest state in the demonstrations to sample the skill via $p(z|s_{\text{closest}})$; and **SPiRL (H-step)** uses the same H-step-ahead look-up approach as FIST and samples a skill based on $p(z|s_{\text{closest}+H})$.

Table 1: Ablation on the impact of semi-parametric approach and the future conditioning of skill prior

| Task (Unseen) | FIST (ours) | FIST (Euc.) | SPiRL | SPiRL (closest) | SPiRL (H-steps) |
|---|---|---|---|---|---|
| Microwave, Kettle, **Top Burner**, Light Switch | $\mathbf{3.6 \pm 0.16}$ | $3.3 \pm 0.15$ | $2.1 \pm 0.48$ | $0.0 \pm 0.0$ | $0.2 \pm 0.13$ |
| **Microwave**, Bottom Burner, Light Switch, Slide Cabinet | $2.3 \pm 0.5$ | $\mathbf{2.8 \pm 0.42}$ | $2.3 \pm 0.49$ | $2.8 \pm 0.44$ | $0.0 \pm 0.0$ |
| Microwave, **Kettle**, Slide Cabinet, Hinge Cabinet | $\mathbf{3.5 \pm 0.3}$ | $3.0 \pm 0.32$ | $1.9 \pm 0.29$ | $0.0 \pm 0.0$ | $0.0 \pm 0.0$ |
| Microwave, Kettle, **Slide Cabinet**, Hinge Cabinet | $\mathbf{4.0 \pm 0.0}$ | $1.3 \pm 0.14$ | $3.3 \pm 0.38$ | $2.4 \pm 0.51$ | $1.5 \pm 0.41$ |

Comparison of SPiRL results suggests that the performance would degrade if the semi-parameteric lookup method is used to pick the conditioning state in SPiRL. This is not surprising since the model in SPiRL is trained to condition on the current state and is unable to pick a good skill $z$ based on $p(z|s_{\text{closest}})$ in SPiRL (closest) or $p(z|s_{\text{closest}+H})$ in SPiRL (H-steps). Therefore, it is crucial to have the prior as $p(z|s_t, s_{t+H})$, so that we can condition on both the current and future goal state.

We also note that FIST (Euc.) results are slightly worse in 3 out of 4 cases. This suggests that (1) contrastive distance is better than Euclidean to some degree (2) the environment's state space is simple enough that Euclidean distance still works to a certain extent. It is worth noting that the Euclidean distance, even with comparable results to the contrastive distance in the kitchen environment, is not a general distance metric to use and would quickly lose advantage when state representation gets complex (e.g. pixel representation).

**The effect of skill pre-training and fine-tuning on FIST**: To adjust the pre-trained skill-set to OOD tasks (e.g. moving the kettle while it is excluded from the skill dataset) FIST requires fine-tuning on the downstream demonstrations. We hypothesize that without fine-tuning, the agent should be able to perfectly imitate the demonstrated sub-trajectories that it has seen during training, but should start drifting away when encountered with an OOD skill. We also hypothesise that pre-training on a large dataset, even if it does not include the downstream demonstration sub-trajectories, is crucial for better generalization. Intuitively, pre-training provides a behavioral prior that is easier to adapt to unseen tasks than a random initialization.

To examine the impact of fine-tuning, we compare FIST with *FIST-no-FT* which directly evaluates the semi-parameteric approach with the model parameters trained on the skill dataset without fine-tuning on the downstream trajectories. To understand the effect of pre-training, we compare FIST with *FIST-no-pretrain* which is not pre-trained on the skill dataset. Instead, we directly train the latent variable and inverse skill dynamics model on the downstream data and perform the semi-parametric evaluation of the FIST algorithm.

From the results in Table 2, we observe that fine-tuning is a critical component for adapting to OOD tasks. The scores on *FIST-no-FT* indicate that the agent is capable of fulfilling the sub-tasks seen during skill pre-training but cannot progress onto unseen tasks without fine-tuning. Based on the scores on *FIST-no-pretrain*, we also find that the pre-training on a rich dataset, even when the downstream task is directly excluded, provides sufficient prior knowledge about the dynamics of the environment and can immensely help with generalization to unseen tasks via fine-tuning.

Table 2: We ablate the use of pre-training on offline data, as well as fine-tuning on downstream demonstrations. FIST-no-FT removes the fine-tuning on downstream demonstration step in FIST, while FIST-no-pretrain trains the skills purely from the given downstream data. Without seeing the subtask, FIST-no-FT is unable to solve the downstream subtask. Trained on only downstream data, FIST-no-pretrain is unable to properly manipulate the robot.

| Task (Unseen) | FIST (ours) | FIST-no-FT | FIST-no-pretrain |
|---|---|---|---|
| Microwave, Kettle, **Top Burner**, Light Switch | $\mathbf{3.6 \pm 0.16}$ | $2.0 \pm 0.0$ | $0.5 \pm 0.16$ |
| **Microwave**, Bottom Burner, Light Switch, Slide Cabinet | $\mathbf{2.3 \pm 0.5}$ | $0.0 \pm 0.0$ | $0.7 \pm 0.15$ |
| Microwave, **Kettle**, Slide Cabinet, Hinge Cabinet | $\mathbf{3.5 \pm 0.3}$ | $1.0 \pm 0.0$ | $0.0 \pm 0.0$ |
| Microwave, Kettle, **Slide Cabinet**, Hinge Cabinet | $\mathbf{4.0 \pm 0.0}$ | $2.0 \pm 0.0$ | $0.8 \pm 0.13$ |

## 5 CONCLUSION

We present FIST, a semi-parametric algorithm for few-shot imitation learning for long-horizon tasks that are unseen during training. We use previously collected trajectories of the agent interacting with the environment to learn a set of skills along with an inverse dynamics model that is then combined with a non-parametric approach to keep the agent from drifting away from the downstream demonstrations. Our approach is able to solve long-horizon challenging tasks in the few-shot setting where other methods fail.

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

# A    IMPLEMENTATION DETAILS

## A.1    DISTANCE FUNCTION

As mentioned in Section 3, we wish to learn an encoding such that our distance metric $d$ is the euclidean distance between the encoded states.

$$d(s, s') = ||h(s) - h(s')||^2 \tag{5}$$

To learn the encoder $h$, we optimize a contrastive loss on encodings of the current and future states along the same trajectory. At each update step we have a batch of trajectories $\tau_1, ..., \tau_B$ where $\tau_i = s_{i,1}, ..., s_{i,H}$. For $s_{i,1}$ the positive sample is $s_{i,H}$ and the negative samples are $s_{j,H}$ where $j \neq i$. We use the InfoNCE Loss van den Oord et al. (2018),

$$\mathcal{L}_q = \log \frac{\exp(q^T W k)}{\exp\left(\sum_{i=0}^{K} \exp(q^T W k_i)\right)}, \tag{6}$$

with query $q = h(s_t)$ as the encoded starting state, and the keys $k = h(s_{t+H})$ as the encoded future states along the $K$ trajectories in the dataset $\mathcal{D}$. By doing so, we are attracting the states that are reachable after H steps and pushing those not reachable far out. This approach has been used successfully in prior work in Liu et al. (2020); Emmons et al. (2020) and we use this method for a generalizable solution for finding the state to condition the future on. This particular choice of distance function is a means to an end for obtaining the future state to condition on and further research is required for investigating similar choices for a robust performance across a diverse set of tasks. We leave this investigation for future work.

## A.2    TRAINING

The training for both skill extraction and fine-tuning were done on a single NVIDIA 2080Ti GPU. Skill extraction takes approximately 3-4 hours, and fine-tuning requires less than 10 minutes. Our codebase builds upon the SPiRL released code and is located at https://anonymous.4open.science/r/fist-C5DF/README.md. Hyperparameters used for training are listed in Table 3.

## A.3    FINE-TUNING

Fine-tuning for both FIST and SPiRL follows the same implementation details. It is done only on $\mathcal{D}^{\text{demo}}$ which includes 10 trajectories of the agent fulfilling the long horizon task. These trajectories are segmented into sub-trajectories of length $H = 10$ first (similar to pre-training), and then we update all the network parameters[2] by minimizing the loss in equation 3, for 50 epochs of the small dataset. The original training was done on 200 epochs of the large and diverse dataset. Everything else, including batch size, learning rate, and the optimizer remain the same between pre-training and fine-tuning. The hyper-parameters for fine-tuning are listed in Table 4.

---

[2]During our initial experimental analysis, we tried a version of FIST that only finetuned the inverse skill dynamics model and that did not show any meaningful results. We hypothesise that without fine-tuning the VAE components, the skill-set of the agent will not include the out of distribution parts of the task.

Table 3: Training Hyperparameters

| Hyperparameter | Value |
|---|---|
| **Contrastive Distance Metric** | |
| Encoder output dim | 32 |
| Encoder Hidden Layers | 128 |
| Encoder # Hidden Layers | 2 |
| Optimizer | Adam($\beta_1 = 0.9$, $\beta_2 = 0.999$, LR=1e-3) |
| **Skill extraction** | |
| Epochs | 200 |
| Batch size | 128 |
| Optimizer | Adam($\beta_1 = 0.9$, $\beta_2 = 0.999$, LR=1e-3) |
| $H$ (sub-trajectory length) | 10 |
| $\beta$ | 5e-4 (Kitchen), 1e-2 (Maze) |
| **Skill Encoder** | |
| dim-$\mathcal{Z}$ in VAE | 128 |
| hidden dim | 128 |
| # LSTM Layers | 1 |
| **Skill Decoder** | |
| hidden dim | 128 |
| # hidden layers | 5 |
| **Inverse Skill Dynamic Model** | |
| hidden dim | 128 |
| # hidden layers | 5 |
| **Fine-tuning** | |
| Epochs | 50 |
| Batch size | 128 |
| Optimizer | Adam($\beta_1 = 0.9$, $\beta_2 = 0.999$, LR=1e-3) |

Table 4: Fine-tuning hyperparameters

| Hyperparameter | Value |
|---|---|
| Epochs | 50 |
| Epoch cycle train | 10 |

## B  ENVIRONMENT AND DATASET DETAILS

In this section we explain the tasks and the procedure for data collection for both pre-training the skills and the downstream fine-tuning in each environment separately. The instruction for downloading the dataset as well as its generation is included in our code repository.

**PointMaze.** In this environment, the task is to navigate a point mass through a maze, from a start to a goal location. The outline of the maze is shown in Figure 3. We train the skills on three different datasets, each

blocking one side of the maze. To test the method's ability to generalize to unseen long-horizon tasks, we use 10 expert demonstrations that start from random places in maze, but end at a goal within the blocked region. This ensures that our demonstrated trajectories are out of distribution compared to training data. We evaluate the performance by measuring the episode length and the success rate in reaching the demonstrated goals. The data for PointMaze is collected using the same scripts provided in the D4RL dataset repository Fu et al. (2020). To generate the pre-training dataset we modified the maze outline to have parts of it blocked. Then we run the same oracle way-point controller on the new maze map and collect continuous trajectories of the agent navigating to random goals, for a total of 4 million transitions. For downstream demonstrations we unblock the blocked region, pick a fixed goal within that region, and have the agent navigate from random reset points to the picked goal location. The observation consists of the (x, y) location and velocities.

**AntMaze.** The task is to control a quadruped ant to run to different parts of the maze. The layout of the maze is the same as PointMaze, and the same sides are blocked off. Similar to PointMaze we measure the episode length and success rate as our evaluation metric. The data for AntMaze is solely based on the "ant-large-diverse-v0" dataset in D4RL. To construct the pre-training data we filter out sub-trajectories that contain states within the blocked region. By doing so, we effectively exclude the region of interest from the pre-training dataset. The result are datasets with 47165, 58329, and 50237 number of transitions, for the Bottom, Right, and Left blocked regions respectively. For the downstream demonstrations we randomly select 10 from the excluded trajectories that start outside the blocked region and end in the blocked region, shown in Figure 3.

**Kitchen.** The task is to use a 7-DoF robotic arm to manipulate different parts of a kitchen environment in a specific order (e.g. open a microwave door or move the kettle)[3]. During skill extraction we pre-process the offline data to exclude interactions with certain objects in the environment (e.g. we exclude interactions with the kettle). However, for the demonstrations we pick four sub-tasks one of which includes the objects that were excluded from the skill dataset (e.g. if the kettle was excluded, we pick the task to be to open the microwave, move the kettle, turn the top burner, and slide the cabinet door). In evaluation, for completion of each sub-task in the order consistent with the downstream demonstrations, the agent is awarded with a reward of 1.0 for a total max reward of 4.0 per episode.

The pool of demonstrations are downloaded from the repository of Gupta et al. (2019) located at `https://github.com/google-research/relay-policy-learning`. There are a total of 24 multi-task long horizon sets of trajectories that the data is collected from. Each trajectory set is sampled at least 10 times via VR tele-operation procedure and the filenames indicate what the agent is trying to achieve (e.g. microwave-kettle-switch-slide). For creating the pre-training data, we filter the trajectory sets, solely based on the filenames that do not include the keyword for the task of interest. This will essentially remove all multi-task trajectories that include the sub-task and not just the part that we do not want. It is also worth noting that the excluded object (e.g. the kettle) is still part of the environment and all its state vectors are still visible to the agent, despite the exclusion of the "interaction" with that object. The dataset size for each case is shown in Table 5.

---

[3]The environment dynamics are still stochastic and therefore, a simple replication of actions would not fulfill the tasks robustly

Table 5: Kitchen pre-training dataset sizes

| Tasks | # of trajectories |
|---|---|
| Microwave, Kettle, **Top Burner**, Light Switch | 285 |
| **Microwave**, Bottom Burner, Light Switch, Slide Cabinet | 236 |
| Microwave, **Kettle**, Slide Cabinet, Hinge Cabinet | 230 |
| Microwave, Kettle, **Slide Cabinet**, Hinge Cabinet | 198 |

# C EXPERIMENTAL RESULTS

## C.1 TABLE OF RESULTS

Table 6: Comparison of our approach to other baselines on the Maze environments. For each experiment we report the average episode length from 10 fixed starting positions with the standard error across 10 evaluation runs (*lower* is better). We also report success rate and its standard deviation. The maximum episode length for PointMaze and AntMaze are 2000 and 1000, respectively.

| | | FIST (Ours) | | SPiRL | | BC+FT | |
|---|---|---|---|---|---|---|---|
| Blocked Region | Environment | Episode Length | Success Rate | Episode Length | Success Rate | Episode Length | Success Rate |
| Left | PointMaze | **363.87 ± 18.73** | 0.99 ± 0.03 | 1966.7 ± 32.54 | 0.02 ± 0.04 | 1089.76 ± 173.74 | 0.74 ± 0.11 |
| Right | PointMaze | **571.21 ± 38.82** | 0.91 ± 0.07 | 2000 ± 0 | 0.0 ± 0.0 | 1918.99 ± 43.65 | 0.07 ± 0.06 |
| Bottom | PointMaze | **359.82 ± 3.62** | 1.0 ± 0.0 | 2000 ± 0 | 0.0 ± 0.0 | 1127.47 ± 148.24 | 0.87 ± 0.10 |
| Left | AntMaze | **764.36 ± 8.93** | 0.32 ± 0.04 | 1000 ± 0 | 0.0 ± 0.0 | 1000 ± 0 | 0.0 ± 0.0 |
| Right | AntMaze | **903.98 ± 12.01** | 0.22 ± 0.12 | 1000 ± 0 | 0.0 ± 0.0 | 1000 ± 0 | 0.0 ± 0.0 |
| Bottom | AntMaze | **923.22 ± 6.36** | 0.21 ± 0.07 | 957.85 ± 8.62 | 0.12 ± 0.07 | 1000 ± 0 | 0.0 ± 0.0 |

Table 7: Comparison of average episode reward for our approach against other baselines on the KitchenRobot environment. The average episode reward (with a max. of 4) along with its standard error is measured across 10 evaluation runs (*higher* is better). Each bolded keyword indicates the task that was excluded during skill data collection.

| Task (Unseen) | Environment | FIST (Ours) | SPiRL | BC+FT |
|---|---|---|---|---|
| Microwave, Kettle, **Top Burner**, Light Switch | KitchenRobot | **3.6 ± 0.16** | 2.1 ± 0.48 | 0.0 ± 0.0 |
| **Microwave**, Bottom Burner, Light Switch, Slide Cabinet | KitchenRobot | **2.3 ± 0.5** | **2.3 ± 0.5** | **2.2 ± 0.28** |
| Microwave, **Kettle**, Slide Cabinet, Hinge Cabinet | KitchenRobot | **3.5 ± 0.3** | 1.9 ± 0.09 | 1.3 ± 0.47 |
| Microwave, Kettle, **Slide Cabinet**, Hinge Cabinet | KitchenRobot | **4.0 ± 0.0** | 3.3 ± 0.38 | 1.0 ± 0.32 |

## C.2 EXTRA ABLATIONS

**Imitation Learning over skills vs. atomic actions.** FIST is comprised of two coupled pieces that are both critical for robust performance: the inverse dynamics model over skills and the non-parametric evaluation algorithm. In this experiment we measure the influence of inverse skill dynamics model $q_\psi(z|s_t, s_{t+H-1})$.

An alternative baseline to learning skill dynamics model is to learn an inverse dynamics model on atomic actions $q_\psi(a_t|s_t, s_{t+H-1})$ and perform goal-conditioned behavioral cloning (Goal-BC). This model outputs the first action $a_t$ required for transitioning from $s_t$ to $s_{t+H-1}$ over $H$ steps. We can combine this model with FIST's non-parametric module to determine the $s_{t+H-1}$ to condition on during the evaluation of the policy. As shown in Table 8, temporal abstraction obtained in learning an inverse skill dynamics model is a critical factor in the performance of FIST.

Table 8: We ablate the use of our inverse skill dynamics model by replacing it with an inverse dynamics model on atomic actions. The baseline ablations only succeed on one out of the four tasks. BC learns an inverse dynamics model that takes in state as input and outputs a distribution over atomic actions. Goal-BC uses both state and the goal (sub-task) as input.

| Task (Unseen) | FIST (ours) | Goal-BC |
|---|---|---|
| Microwave, Kettle, **Top Burner**, Light Switch | **3.6 ± 0.16** | 0.0 ± 0.0 |
| **Microwave**, Bottom Burner, Light Switch, Slide Cabinet | **2.3 ± 0.5** | 1.2 ± 0.3 |
| Microwave, **Kettle**, Slide Cabinet, Hinge Cabinet | **3.5 ± 0.3** | 1.8 ± 0.44 |
| Microwave, Kettle, **Slide Cabinet**, Hinge Cabinet | **4.0 ± 0.0** | 0.9 ± 0.1 |

**Is our contrastive distance function an optimal approach for picking the future state to condition on?**
For environments such as PointMaze, where we have access to the same waypoint controller that generates the demonstrations, we can use the ground truth environment dynamics to calculate the oracle state that the waypoint controller would be at, H steps in the future, and use that as the foal. This waypoint controller is not available for the kitchen environment, since the demonstrations were collected using VR tele-operation.

In Table 9, FIST (Oracle) uses the waypoint controller oracle look-up. The results show that with a better future conditioning, an oracle approach can solve the pointmaze task even better than FIST with the contrastive distance. Improving the current semi-parametric approach for obtaining the future conditioning state could be a very interesting direction for future work.

Table 9: We ablate FIST against an oracle version on pointmaze which uses the ground truth way point planner that has access to the exact state that the agent will end up at H-steps in the future (if it commits to the optimal path).

| Section | FIST | | FIST (oracle) | |
|---|---|---|---|---|
| | Episode Length | Success Rate | Episode Length | Success Rate |
| Left | 363.87 ± 18.73 | 0.99 ± 0.03 | 236.00 ± 1.02 | 1.0 ± 0.00 |
| Right | 571.21 ± 38.82 | 0.91 ± 0.07 | 280.93 ± 5.61 | 1.0 ± 0.00 |
| Bottom | 359.82 ± 3.62 | 1.0 ± 0.00 | 269.89 ± 3.75 | 1.0 ± 0.00 |

**One-shot Imitation Learning**: The FIST algorithm can be directly evaluated on one-shot in-distribution downstream tasks without any fine-tuning. In this experiment, we want to see if the agent can pick up the right mode within its skill-set with only one demonstration for fulfilling a long-horizon task in the kitchen environment. The difference between this experiment and our main result is that the down-stream task is within the distribution of its pre-trained skill-set. This is still a challenging task since the agent needs to correctly identify the desired mode of skills.

Our hypothesis is that in SPiRL, the skill prior is only conditioned on the current state and therefore is, by definition, a multi-modal distribution and would require more data to adapt to a specific long-horizon trajectory. For instance, in the kitchen environment, after opening the microwave door, the interaction with any other objects in the environment is a possible choice of skills that can be invoked. However, in FIST, by conditioning the skill prior on the future states, we fit a uni-modal distribution over skills. In principle, there should be no need for fine-tuning for invoking those skills within the distribution of the pre-trained skill set.

We compare our approach to SPiRL (Section 4.2) as a baseline. In addition, we can provide supervision on which skills to invoke to fulfill the long-horizon task by fine-tuning SPiRL (hence *SPiRL-FT*) for a few epochs on the downstream demonstration. As summarized in Table 10, FIST, without any fine-tuning, can fulfill all the long-horizon tasks listed with almost no drift from the expert demonstration. We also see that

it is tricky to fine-tune SPiRL in a one-shot setting, as fine-tuning only on one demonstration may cause over-fitting and degradation of performance.

Table 10: With all subtasks seen in the skill dataset, FIST is able to imitate a long-horizon task in the kitchen environment. We compare to a baseline method, SPiRL, which fails to follow the single demo.

| Order of tasks (seen in the skill dataset) | FIST (ours) | SPiRL-FT | SPiRL-no-FT |
|---|---|---|---|
| Kettle, Bottom Burner, Slide Cabinet, Hinge Cabinet | $\mathbf{4.0 \pm 0.0}$ | $0.8 \pm 0.19$ | $2.4 \pm 0.35$ |
| Kettle, Top Burner, Light Switch, Slide Cabinet | $\mathbf{3.8 \pm 0.19}$ | $0.5 \pm 0.16$ | $1.1 \pm 0.22$ |
| Microwave, Kettle, Slide Cabinet, Hinge Cabinet | $\mathbf{4.0 \pm 0.0}$ | $1.1 \pm 0.22$ | $1.0 \pm 0.37$ |
| Top Burner, Bottom Burner, Slide Cabinet, Hinge Cabinet | $\mathbf{4.0 \pm 0.0}$ | $0.1 \pm 0.1$ | $0.6 \pm 0.25$ |

## D  BROADER IMPACTS AND LIMITATIONS

**Limitations**   As with all imitation learning methods, the performance of FIST is related to the quality of the provided demonstrations. Concretely, when the skill training demonstrations are poor, we expect the extracted skills to be also sub-optimal, thus, hurting downstream imitation performance. To better understand this limitation, we analyze an extremely noisy versions of the PointMaze dataset and use it for skill extraction. As shown in Table 11, despite achieving a high success rate, the episode length is substantially worse than FIST trained on expert data.

Learning structured skills from noisy offline data is an exciting direction for future research.

| Environment | Episode Length | Success Rate |
|---|---|---|
| PointMaze | $621.02 \pm 69.87$ | $1.0 \pm 0.0$ |

Table 11: We evaluate FIST on the maze environment with goal at the bottom when the inverse skill model is trained on an extremely noisy dataset. In this case, FIST achieves sub-optimal performance, or is unable to imitate the test time demonstration.

**Broader Impacts**   The ability to extract skills from offline data and adapt them to solve new challenging tasks in few-shot could be impactful in domains where large offline datasets are available but control is challenging and cannot be manually scripted. Examples of such domains include autonomous vehicle navigation, warehouse robotics, digital assistants and perhaps in the future, home robots. However, there are also negative potential consequences. First, since in real-world settings offline data will be collected from users at scale there will likely be privacy concerns, especially for video data collected from users' cars or homes. Additionally, since FIST extracts skills, without labeled data, quality for large datasets becomes increasingly opaque and if there are harmful skills or behavior present in the dataset FIST may extract those and use them during deployment which could have unintended consequences. A promising direction for future work is to include a human in the loop for skill verification.

