# OpenReview forum: "Hierarchical Few-Shot Imitation with Skill Transition Models"
_ICLR.cc/2022/Conference — ICLR 2022 Poster_

### Official Review · Reviewer_JEuK · 2021-11-02

**Correctness:** 3
**Technical Novelty And Significance:** 2
**Empirical Novelty And Significance:** 3
**Recommendation:** 6
**Confidence:** 3

**Main Review:**

*Strengths*

The method proposes some simple extensions to the method in SPiRL that seem to be effective. It makes sense that having the skill grounding take not just a current state but a goal state should make mapping to the correct skill easier. Adding in finetuning on the few demos is also sensible idea. Lastly, using a learned temporal distance to query the closest state in the provided demo makes sense, and the results indicate it performs better than using Euclidean distance.

The experimental results show that the method seems to perform SPiRL significantly across a range of environments.

*Weaknesses*

I have some questions about the justification for the performance gap between this method and SPiRL. First, to confirm - SPiRL is also finetuned on the few shot demos correct? So when the paper states that SPiRL "The skills learned by SPiRL fail to generalize when the point mass falls outside of the training distribution, causing it to get stuck in corners", why is this an issue for SPiRL but not FIST when both are finetuned?

In general I think it makes sense that skill selection should be conditioned on not just the current state but a goal state H steps ahead. However from what I can tell this is the primary difference between FIST and SPiRL, is it the case that just this change is the difference between the 0% -> 80% success between SPiRL and FIST? It would be great if the authors could expand more on why exactly SPiRL fails, and why the change to having the skill prior take s_{t+H} impacts performance so much. It seems like a minor change but the results indicate that its important.

There are a few typos and grammatical errors throughout the paper, and in general the presentation/figures could be improved. Fig 4 should have some measure of uncertainty/standard error. I also can't find how many trials/seeds are used in the experiments.

**Summary Of The Paper:**

The paper presents an approach for skill extraction from an offline dataset of demos/interaction data, for use in few shot imitation learning of previously unseen long horizon tasks. The method trains a skill encoder mapping trajectories to a latent skill space, as well as a decoder which outputs actions conditioned on a state and skill (similar to SPiRL, Pertsch et al). The method also learns a skill prior which maps a current state and goal state (H steps ahead of current state) to the latent skill space. The main different to SPiRLis that the skill prior takes as input both a current and goal state, while in SPiRL it took in only takes in the current state. The skill prior and skill encoder/decoder are jointly trained on the demos.

Then during evaluation of a new long horizon task given a few demos, the agent takes in as input the current state, queries a nearby state in the provided demos, recover the state H steps ahead, and apply the skill prior given the s, and s_{t+H}. The querying nearby states is using a learned distance function trained to capture temporal distance. All components of the method are finetuned on the few demos provided for the new task.

Experiments suggest this method outperforms SPiRL and naive BC + finetuning on unseen multi step tasks in a simulated Franka kitchen environment and on ant maze.

**Summary Of The Review:**

Overall the paper presents some technical extensions over SPiRL, and seems to have significant performance improvements. I have some questions about the experiments, but if addressed I think the papers results are solid.

---

### Official Review · Reviewer_s3dr · 2021-11-03

**Correctness:** 4
**Technical Novelty And Significance:** 2
**Empirical Novelty And Significance:** 3
**Recommendation:** 6
**Confidence:** 4

**Main Review:**

Overall, this paper is quite interesting to read and the presented problem is worth being investigated. I will write its strength and weakness respectively.

[Major concern:]
- As I summarized above, the first contribution on long-horizon task learning by extracting behaviour skills and by inverse skill model to selecting skills seems trivial since the basic components were proposed in previous work (Pertsch et al. 2020, Ajay et al. 2021).
- The second contribution I think is more interesting but still lacks in-depth investigation.
  - For example, the skill level is defined on state-action sequences, but the learned distance function for finding similar states is based on the state level. Will there be any issues? Any distributional perspective to explain why your distance function can make the learned skills generalize or adapts?

[Minor comments:]
- It seems to me this paper involves lots of work there, however, the main draft is underdeveloped that it even lacks a conclusion section and a Reference section name. The format needs to be double-checked.
- Could the authors explain the details in the process "LookAhead" in Algorithm 1?

**Summary Of The Paper:**

This paper tackles the problem of learning generalizable long-horizon tasks from offline human demonstrations with the below two insights:

- (1) The long-horizon task learning is enabled by extracting behaviour priors Z as basic skills from offline data, which has been seen in previous publications  (Pertsch et al. 2020).
- (2) The generalization is enabled by matching the new states from adapted envs with the states from demonstrations using a pre-trained distance function, which is common in few-shot imitation learning.

Experiments on a simulated navigation task (Pointmass and Ant) and a robotic manipulation task (Kitchen) were performed to prove the effectiveness of the proposed approach.

======================================================

Clarifications:

Since task generalization benchmark is not well established in our research community currently, I feel it is worthy to state clearly what is the claimed generalization here. Specifically, the demonstrated generalization performance is under the few-shot learning setting. For example,

- (1) In Pointmass and Ant env, some regions are blocked in offline human demonstrations.
    - When testing generalization, the authors provide a few more human demonstrations covering the blocked regions. The proposed method will thus generalize.
- (2) In the Kitchen task, some objects are removed from offline human demonstrations.
    - When testing generalization, the authors provide a few more human demonstrations covering the removed objects.


**Summary Of The Review:**

Overall, it is a good paper to read and the authors have done a lot of works. However, the major contribution is not sufficiently novel to be clearly accepted in ICLR. As a result, I recommend a weak acceptance.

---

### Official Review · Reviewer_LjxN · 2021-11-04

**Correctness:** 3
**Technical Novelty And Significance:** 3
**Empirical Novelty And Significance:** 3
**Recommendation:** 8
**Confidence:** 5

**Main Review:**

### Strengths

- This paper tackles an important problem of solving a new task without a huge number of interactions utilizing skills and a few downstream demonstrations.
- The proposed method is intuitive and shows potential for a versatile agent which can quickly adapt to a new task.
- The ablation study dissects components of the proposed method and shows the importance of each of the components, supporting the paper's claims.

### Weaknesses

- In the kitchen environment, all tasks start from Microwave. In some of the tasks, the baseline approaches (Goal-BC, FIST-no-pretrain, SPiRL (closest), SPiRL (H-steps)) fail to accomplish the Microwave sub-task for all random seeds while in some other cases, they succeed. How can we interpret these results?

- In Algorithm 1, a skill $z$ is sampled every environment step. Is this better than executing $H$ actions from the sampled skill $z$? To adjust to the given few-shot demonstrations, inferring a lookahead state and skill every time step makes sense. But, in terms of temporally consistent action execution, executing a skill as a whole may work better.


### Minor comments

- In Section 3.2 (iii), $z \sim q_\psi(z|s_t,s^*_{t})$ should be $z \sim q_\psi(z|s_t,s^*_{t+H})$.


**Summary Of The Paper:**

This paper proposes a novel semi-parametric way of few-shot imitation learning. It assumes access to a large task-agnostic dataset to extract fixed-length skills $\pi(a|s,z)$ and an inverse skill dynamics model $p(z|s_t, s_{t+H})$, which infers required skills given the current and future states. With the inverse skill dynamics model, an agent can sample a skill to reach a given goal state. When tackling a new downstream task with a few demonstrations, this method seeks for the closest state $s^*_{t}$ in these demonstrations and sets $s^*_{t+H}$ as a new goal. Since we assume $s_t$ and $s^*_{t}$ are similar, reaching $s^*_{t+H}$ should be plausible with the learned skills. The experiments show that this approach successfully solves unseen tasks with a few demonstrations and outperforms the behavioral cloning and skill prior RL baselines.

**Summary Of The Review:**

Overall, this paper proposes an intuitive and novel approach for skill-based few-shot imitation learning. The experimental results show the advantage of learning the inverse skill dynamics and the proposed semi-parametric imitation learning approach.

---

### Official Review · Reviewer_enSE · 2021-11-06

**Correctness:** 3
**Technical Novelty And Significance:** 3
**Empirical Novelty And Significance:** 2
**Recommendation:** 6
**Confidence:** 5

**Main Review:**

Strengths
-------------

- I like the inverse skill dynamics model idea
- Nice implementation of skill learning + fine-tuning
- Thorough analysis, ablations and results


Weaknesses
------------------

- The learned distance function feels like it is "forced" because of the inverse skills model and not thought out in detail. Although the experiments/analysis in the supplementary reduced this feeling a lot. This is not a big issue, but if the authors can shed more light to this, it'll make the paper better.
- The algorithm uses very big offline datasets (on unstructured data) and at least 10 full demonstrations of successfully completing the task. Here I have 2 concerns:
    - The success rate even after these big datasets is relatively low (<80% on average). Couldn't we do better?
    - What is practical usage of the proposed method? If I have access to big datasets, I think I can do many other more effective things. For example, insert priors in the form of structures in the policies (see {1,2,3}). Of course I understand that here we do not use any prior at all, but I'd like to see at least a discussion on this.
- The related work section and motivation is quite narrow. Overall the motivation for developing such a method is rather weak. In other words, I see limited impact. A few examples:
    - Imitation Learning: the field is vast and has been tackled from many different perspectives. Perhaps the biggest literature for imitation learning comes from the robotics community (see {3,4,5}). The related work section/motivation needs quite some effort to improve.
   - Skill Extraction with Behavioral Priors and Few-Shot Learning: here again the evolutionary-based perspective is not even mentioned. There are many approaches that compute behavioral repertoires to enable fast (or few-shot) adaptation (see {3,6,7}). Here I'd also like to see some discussion.
- The supplementary material needs to better describe the datasets (samples of offline data, a few visual examples, ...)
- The supplementary video does not show ANY video of the learned behaviors!!


References
----------------

{1}: Bahl, S., Mukadam, M., Gupta, A. and Pathak, D., 2020. Neural dynamic policies for end-to-end sensorimotor learning. NeurIPS.

{2}: Stulp, F. and Sigaud, O., 2013. Robot skill learning: From reinforcement learning to evolution strategies. Paladyn, Journal of Behavioral Robotics, 4(1), pp.49-61.

{3}: Chatzilygeroudis, K., Vassiliades, V., Stulp, F., Calinon, S. and Mouret, J.B., 2019. A survey on policy search algorithms for learning robot controllers in a handful of trials. IEEE Transactions on Robotics, 36(2), pp.328-347.

{4}: Billard, A., Calinon, S., Dillmann, R. and Schaal, S., 2008. Survey: Robot programming by demonstration (No. BOOK_CHAP, pp. 1371-1394). Springrer.

{5}: Osa, T., Pajarinen, J., Neumann, G., Bagnell, J.A., Abbeel, P. and Peters, J., 2018. An algorithmic perspective on imitation learning. Foundations and Trends® in Robotics.

{6}: Cully, A., Clune, J., Tarapore, D. and Mouret, J.B., 2015. Robots that can adapt like animals. Nature, 521(7553), pp.503-507.

{7}: Pautrat, R., Chatzilygeroudis, K. and Mouret, J.B., 2018, May. Bayesian optimization with automatic prior selection for data-efficient direct policy search. In 2018 IEEE International Conference on Robotics and Automation (ICRA) (pp. 7571-7578). IEEE.


**Summary Of The Paper:**

The paper presents an approach for skill learning and fine-tuning. The main idea of the manuscript is to combine offline skill learning with few online demonstrations to provide efficient learning experience in out of distribution (OOD) tasks. There are two main novelties in the paper:

- Learning of an inverse skill dynamics model that infers which skills should be used given the current state and a future state
- Learning a distance function between states that can be used to select states from the demos as "target/future" states for better generalization

Apart from the technical novelties, the paper presents ablation studies with intuitions on which of the parts of the new approach (FIST) are important and why.

**Summary Of The Review:**

The paper was an interesting and nice read. Skill learning and fast adaptation are key features that can enable widespread adoption of robots. I have however three main concerns: (a) the algorithm uses very big offline datasets + 10 full demonstrations (states and actions) and success rate of completing the tasks is relatively low (<80% on average); we can certainly do better than this with minor prior information, (b) the related work/motivation part needs more work (it is not convincing me at the moment) and (c) the supplementary video does not show the performance of the algorithm.

---

### Decision · Program_Chairs · 2022-01-20

**Decision:**

Accept (Poster)

**Comment:**

The reviewers agree that addressing long-horizon tasks with off-line learning and fine tuning afterwards from demonstrations is an interesting and relevant topic. The technical ideas about learning a relevance metric to select relevant off-line data, and to learn an inverse skill dynamics models. The experimental results are convincing, even if success rates are sometimes lower than expected. All reviewers recommend acceptance of the paper.